# The Effect of Genetic Polymorphism in Response to Body Weight Reduction in Japanese Patients with Nonalcoholic Fatty Liver Disease

**DOI:** 10.3390/genes12050628

**Published:** 2021-04-22

**Authors:** Yuya Seko, Kanji Yamaguchi, Nozomi Tochiki, Kota Yano, Aya Takahashi, Shinya Okishio, Seita Kataoka, Keiichiroh Okuda, Atsushi Umemura, Michihisa Moriguchi, Yoshito Itoh

**Affiliations:** Department of Molecular Gastroenterology and Hepatology, Graduate School of Medical Science, Kyoto Prefectural University of Medicine, Kyoto 602-8566, Japan; yuyaseko@koto.kpu-m.ac.jp (Y.S.); ykanji@koto.kpu-m.ac.jp (K.Y.); ntochiki@koto.kpu-m.ac.jp (N.T.); yanokota@koto.kpu-m.ac.jp (K.Y.); ayataka@koto.kpu-m.ac.jp (A.T.); okishin@koto.kpu-m.ac.jp (S.O.); s1120@koto.kpu-m.ac.jp (S.K.); k-okuda@koto.kpu-m.ac.jp (K.O.); aumemura@koto.kpu-m.ac.jp (A.U.); mmori@koto.kpu-m.ac.jp (M.M.)

**Keywords:** NAFLD, *PNPLA3*, *HSD17B13*, SNP, diet therapy

## Abstract

Background: weight loss as a result of lifestyle intervention is effective when treating non-alcoholic fatty liver disease (NAFLD). We estimated the effects of *PNPLA3* rs738409 and *HSD17B13* rs6834314 variants in response to diet therapy in Japanese patients with NAFLD. Methods: we analyzed the correlation between the change in liver stiffness and change in body weight in 140 patients administered diet therapy for 1-year, according to *PNPLA3* and *HSD17B13* genotypes. Results: the bodyweight (BW) reduction rate was greater in patients with the *PNPLA3* genotype CC than CG and GG (*p* = 0.035). Change in liver stiffness measurement (LSM) was significantly associated with a change in BW in *PNPLA3* CG/GG (r = 0.279/0.381), but not in *PNPLA3* CC (*p* = 0.187). Change in LSM was correlated with change in BW only in patients with *HSD17B13* AG/GG (r = 0.425), but not the AA genotype (*p* = 0.069). A multivariate analysis identified that a change in LSM was correlated with a change in BW in carriers of *HSD17B13* AG/GG (B = 3.043, *p* = 0.032), but not *HSD17B13* AA. The change in LSM of patients with a BW reduction of more than 7% (0.50) was significantly greater than that of patients with a BW reduction of less than 7% (0.83) (*p* = 0.038). Conclusions: in Japanese patients with NAFLD, *HSD17B13* rs6834314 polymorphism is associated with the change in LSM by lifestyle intervention. The approach, including genetic assessments, may contribute to the establishment of appropriate therapeutic strategies to treat NAFLD.

## 1. Introduction

Non-alcoholic fatty liver disease (NAFLD) and non-alcoholic steatohepatitis (NASH) are hepatic phenotypes of metabolic syndrome, which are common causes of chronic liver disease and, ultimately, liver cirrhosis, hepatic failure, and hepatocellular carcinoma (HCC), in the absence of significant alcohol consumption [1,2]. A large cohort study identified hepatic fibrosis as the only predictive factor among those evaluated for liver-related events and mortality in patients with NAFLD [3]. Bodyweight (BW) reduction by lifestyle intervention (e.g., calorie restriction or exercise) is the most effective management option for NAFLD. It ameliorates aminotransferases and improves histological severity, including hepatic fibrosis [4,5,6]. Previous studies reported that BW reduction of more than 10% from baseline could induce NASH resolution and fibrosis regression in most NAFLD patients. However, BW reduction of 5% can only improve histological severity in 35–45% of NAFLD patients [7].

Genetic factors are also involved in the histological progression in individuals with liver diseases. The single-nucleotide polymorphism (SNP) rs738409 in the patatin-like phospholipase domain containing 3 (*PNPLA3*) gene is associated with NAFLD and NASH [8,9,10]. We previously reported in a longitudinal study that advanced fibrosis and the *PNPLA3* rs738409 GG genotype are predictive factors for HCC development in Japanese patients with NAFLD [11]. The mechanism underlying the effects of *PNPLA3* on hepatic fibrosis and hepatocarcinogenesis is still unclear. Plausible mechanisms include the effects of *PNPLA3* on lipid droplet remodeling, very-low-density lipoprotein secretion, retinol metabolism via retinyl-palmitate lipase activity, and the mRNA level of Fas ligand [12]. Recently, Abul-Husn et al. reported that a splice variant (rs72613567) of the 17-β-hydroxysteroid dehydrogenase 13 gene (*HSD17B13*) has a protective effect on the development of chronic liver disease [13]. Another study investigated the association between *HSD17B13* and liver disease and showed that the *HSD17B13* rs72613567 TA variant reduced the risks of alcohol-, NAFLD-, and hepatitis C-related cirrhosis and alcohol-related HCC [14]. In NAFLD subjects, the minor allele rs6834314 of *HSD17B13* was associated with increased steatosis, but decreased inflammation and ballooning via its hepatic retinol dehydrogenase activity [15]. Furthermore, the carriage of this variant attenuated the risk of developing liver injury conferred by *PNPLA3* polymorphisms. We recently reported that carriage of the *HSD17B13* rs6834314 G allele attenuated the effect of the *PNPLA3* rs738409 GG genotype on advanced hepatic fibrosis [16].

The genetic variants were associated not only with disease progression, but also with response to therapy. In a pilot study by Sevastianova et al., it was reported that patients with *PNPLA3* genotype GG showed greater reduction of intrahepatic triglyceride content (IHTG) than patients with genotype CC by a short period hypocaloric diet intervention [17]. Shen et al. also reported that the presence of G allele was associated with greater reduction in IHTG, body weight, waist-to-hip ratio, total cholesterol, and low-density lipoprotein (LDL) cholesterol by a 1-year lifestyle modification [18]. As we mentioned, the *PNPLA3* and *HSD17B13* genotypes are in a mutual relationship in pathology of NAFLD, and improvement of liver fibrosis is more important than that of steatosis. Thus, an investigation of the effects of these polymorphisms in response to lifestyle intervention in Japanese patients is needed.

In this study, we adopted vibration-controlled transient elastography (VCTE) for evaluating liver stiffness measurement (LSM). Because of its non-invasive aspects and convenience, VCTE is frequently used in clinical settings for fibrosis staging in NAFLD [19]. VCTE is a simple and readily available examination to carry out, but obesity can be a reason for failed VCTE. Giuffrè M, et al. reported that skin-to-liver distance affected the LSM in obese patients. We should take into consideration skin-to-liver distance when interpreting LSM in NAFLD [20].

The aims of this study were to (1) identify the association between carriage of *PNPLA3* rs738409 genotype and change in bodyweight (BW) and LSM, and (2) determine whether such *HSD17B13* rs6834314 genotype influences the effect of the *PNPLA3* rs738409 G variant, concerning the response to diet therapy in Japanese patients with biopsy-proven NAFLD.

## 2. Materials and Methods

### 2.1. Patients

A total of 283 Japanese patients who were diagnosed with NAFLD by liver biopsy at the Department of Gastroenterology and Hepatology, Kyoto Prefectural University of Medicine (Kyoto, Japan), from January 2002 to March 2019, were enrolled in this study. 0We diagnosed NAFLD based on the liver biopsy finding of steatosis in ≥5% of hepatocytes and the exclusion of other liver diseases, including viral hepatitis, autoimmune hepatitis, and drug-induced liver disease. Patients with a daily alcohol consumption >30 g for men and >20 g for women were excluded. This study was approved by the Ethical Review Board of Kyoto Prefectural University of Medicine (ERB-C-1416). All patients provided written informed consent at the time of liver biopsy, and the study was conducted in accordance with the Declaration of Helsinki (2013). 

### 2.2. Physical Examination, Laboratory, and Clinical Parameters

Laboratory assays included blood cell counts and measurements of serum concentrations of albumin, aspartate aminotransferase (AST), alanine aminotransferase (ALT), γ glutamyl transpeptidase, total cholesterol, triglycerides (TG), high-density lipoprotein (HDL) cholesterol, LDL cholesterol, fasting plasma glucose, and hemoglobin A1c. These parameters were measured using standard clinical chemical laboratory techniques. Body mass index (BMI) was calculated as weight in kilograms/(height in meters)2. The Fibrosis-4 (FIB-4) index was calculated as follows: ([age (years) × AST (IU/L)]/platelet count [109/L]) × (ALT [IU/L])1/2. FIB-4 index was reported associated with LSM [21]. Patients were examined in a fasting state at rest in the supine position. The FibroScan (EchoSens, Paris, France) standard M probe was placed on the skin in an intercostal space and LSM was carried out. The operators were experienced gastroenterologists who were conversant with transient elastography. Valid LSM was defined as follows: (i) at least 10 valid measurements with >60% success rate; and (ii) <30% of interquartile range. At the same time, the controlled attenuation parameter (CAP) was measured, only when LSM was valid. 

Patients taking oral hypoglycemic agents and those with a fasting glucose concentration >126 mg/dL or a random glucose concentration >200 mg/dL were diagnosed with diabetes. Patients with serum levels of low-density lipoprotein cholesterol >140 mg/dL, TG >150 mg/dL, and/or high-density lipoprotein cholesterol <40 mg/dL were diagnosed with dyslipidemia. 

### 2.3. DNA Preparation and SNP Genotyping

Genomic DNA was extracted from blood samples using the DNeasy Blood and Tissue kit (Qiagen, Tokyo, Japan). The SNPs rs738409 and rs6834314 were genotyped in each sample using TaqMan SNP genotyping assays (Applied Biosystems, Foster City, CA, USA) with commercially available predesigned SNP-specific primers for PCR amplification and extension reactions, according to the manufacturer’s protocol. 

### 2.4. Liver Histology

The liver biopsy specimens were stained with hematoxylin and eosin, and Masson trichrome stain. The specimens were evaluated by two hepatic pathologists who were blinded to the clinical findings. An adequate liver biopsy sample was defined as that with a length >1.5 cm and/or more than 11 portal tracts. The steatosis severities of <5, 5–33, >33–66, and >66% were assigned steatosis scores of 0, 1, 2, and 3, respectively. Lobular inflammation grades 1, 2, and 3, and ballooning scores of 0, 1, and 2, were defined as mild, moderate, and severe, respectively. The NAFLD activity score was calculated as the sum of the steatosis, lobular inflammation, and hepatocellular ballooning scores. The severity of hepatic fibrosis was staged as follows: stage 1, zone 3 perisinusoidal fibrosis; stage 2, zone 3 perisinusoidal and portal fibrosis; stage 3, zone 3 perisinusoidal, portal, and bridging fibrosis; and stage 4, cirrhosis [22,23,24]. In this study, hepatic stages 3 and 4 were defined as advanced fibrosis.

### 2.5. Statistical Analysis

The categorical clinical data between groups were compared by the chi-squared test or the linear-by-linear trend test; quantitative variables were analyzed using paired t-test, and one-way ANOVA, as appropriate. All changes between baseline and 1 year after were analyzed using two-tailed paired t-tests. The correlation analysis was carried out with Spearman’s correlation analysis. The correlation coefficient was determined; weak, 0.00- 0.39; moderate, 0.40–0.59; strong, 0.60–1.0. Linear regression was used to determine the independent factors associated with quantitative change in LSM. The statistical analyses were performed using SPSS version 22 (SPSS Inc., Chicago, IL, USA). All *p*-values less than 0.05 by two-tailed tests were considered statistically significant.

## 3. Results

### 3.1. Patient Characteristics

Table 1 and Table 2 summarize the demographic profiles and laboratory and histologic data of the 140 patients in this study, according to the *PNPLA3* or *HSD17B13* genotype. The 140 patients, comprising 69 (49.3%) females, had a median age of 55 years and median BMI of 27.6 kg/m2. Of the 140 patients, 52 (37.1%), 45 (32.1%), 17 (12.1%), 17 (12.1%), and 9 (6.4%) were hepatic fibrosis stages 0, 1, 2, 3, and 4 (cirrhosis), respectively. In this cohort, the frequencies of the *PNPLA3* rs738409 genotypes were CC in 30 (21.4%) patients, CG in 58 (41.4%), and GG in 52 (37.1%). The frequencies of the *HSD17B13* rs6834314 genotypes were 54.1% AA in 72 (51.4%), AG in 57 (40.7%), and GG in 11(7.9%), respectively. Because the number of patients with GG were small, we combined AG and GG in this study. The serum levels of TG and HDL cholesterol were significantly different according to the *PNPLA3* genotype (Table 1). The serum levels of AST, ALT, and FIB-4 index were significantly lower in patients with the *HSD17B13* genotype AG/GG compared with genotype AA (Table 2).

### 3.2. Changes in Parameters from Baseline to after 1 Year Diet Therapy and Associations with the PNPLA3 and HSD17B13 Genotypes

After 1 year of diet therapy, BMI, serum levels of AST, ALT, and GGT, CAP, and LSM were significantly reduced (Table 3). Figure 1 showed change in BW (ΔBW), in LSM (ΔLSM), according to *PNPLA3* and *HSD17B13* polymorphism. The *PNPLA3* genotype was associated with a greater reduction rate of BW (CC: 2.9%, CG: 1.3%, GG: 0.2%) (*p* = 0.035). *HSD17B13* polymorphism had no impact on ΔBW (*p* = 0.687). The ΔLSM was not associated with both *PNPLA3* and *HSD17B13* polymorphism (*p* = 0.439, *p* = 0.608, respectively). The ΔLSM had no difference in each fibrosis stage, in both *PNPLA3* and *HSD17B13* polymorphisms.

Figure 2 shows the associations between ΔBW and ΔLSM according to the *PNPLA3* genotype. The ΔLSM had significant positive correlation with ΔBW in *PNPLA3* genotype CG (r = 0.279, *p* = 0.034) and GG (r = 0.381, *p* = 0.009). In contrast, there was no correlation between ΔLSM and ΔBW in genotype CC (*p* = 0.187). The ΔLSM had significant positive correlation with ΔBW in *HSD17B13* genotype AG/GG (r = 0.425, *p* < 0.01), but not in AA (*p* = 0.069) (Figure 3). 

### 3.3. Factors Associated with Change in LSM According to HSD17B13 Polymorphism 

To estimate the factors, which effect ΔLSM due to *HSD17B13* genotypes, we examined the correlations of ΔLSM with ΔALT and ΔBW in patients with *HSD17B13* AA and AG/GG genotypes (Figure 3). In patients with *HSD17B13* AA genotype, ΔLSM was associated with ΔALT (r = 0.318, *p* = 0.009), but not with ΔBW (*p* = 0.069) (Figure 3a). On the other hand, ΔLSM was significantly correlated with both ΔALT (r = 0.339, *p* = 0.006) and ΔBW (r = 0.425, *p* < 0.001) (Figure 3b). We performed multivariate linear regression analysis to evaluate the associations of change in BW, ALT level, AST level, GGT level, TC level, TG level, and *PNPLA3* genotype with change in LSM to the *HSD17B13* genotype (Table 4). Among patients with the *HSD17B13* AA genotype, both changes in ALT and BW did not correlate with change in LSM. On the other hand, among patients with the *HSD17B13* AG/GG genotype, change in BW (per 1 kg; β 3.043, 95% CI 0.276–5.811, *p* = 0.032) was independently associated with change in LSM. The change in LSM in patients with a reduction of BW more than 5% (0.80) and 7% (0.71) were not different from those in patients without achieving it (0.89, 0.89, respectively) (Figure 4a). Among patients with the *HSD17B13* AG/GG genotype, the change in LSM of patients who achieved a BW reduction of more than 5% (0.50) was significantly greater than that of patients with a BW reduction of less than 5% (0.84) (*p* = 0.008). Furthermore, the change in LSM of patients with a BW reduction of more than 7% (0.50) was significantly greater than that of patients with a BW reduction of less than 7% (0.83) (*p* = 0.038) (Figure 4b).

## 4. Discussion

Lifestyle modifications, especially restrictions on calorie intake, are recommended as the most important types of therapy by the NAFLD practice guidelines [25]. Several trials examined the association between BW reduction and improvement of histological features, including fibrosis. In a meta-analysis, patients who lost at least 5% of BW had improved hepatic steatosis; those with more than 7% of BW reduction achieved NAFLD activity score improvement, and BW loss of more than 10% was associated with improvement in all features of NASH [26,27]. However, only 50% of patients were able to achieve at least 7% BW loss in that study. The *PNPLA3* rs738409 G allele is widely known to be associated with more severe NAFLD. At the same time, it is also associated with a greater reduction in hepatic fat content by diet therapy [4,17,18], pharmacotherapy with the DPP-4 inhibitor [28], and bariatric surgery [29,30]. Shen et al. conducted a randomized control study to estimate the effects of *PNPLA3* polymorphism in treating lifestyle modifications in NAFLD [18]. They reported that NAFLD patients carrying the G allele of *PNPLA3* rs738409 demonstrated a greater reduction of BW and IHTG compared to those with the C allele. In the present study, the reduction of BW was greater in patients with *PNPLA3* genotype CC than in patients with genotype CG/GG. Although the reduction of BW was greater according to the predominance of C allele, among patients with BW loss of more than 5%, the reduction of LSM was significantly greater according to the predominance of G allele, GG (0.46), CG (0.82), and CC (0.79) (*p* = 0.021). This discrepancy in BW change may be based on the difference in the restriction of caloric intake. In the previous study, authors provided an individual menu plan consisting of moderate carbohydrates, low fat, low-glycemic index, and low-calorific products with fruits and vegetables in appropriate portions. The patients attended dietary consultation sessions weekly during the first 4 months, and monthly in the following 8 months. We gave a dietary consultation session at baseline. This difference led to a greater BW reduction in a previous study than our study. The higher prevalence of patients with advanced fibrosis in this study may be another reason of the difference between two studies. Our study includes 18.6% of patients with advanced fibrosis, which is greater than the previous study. The reduction in CAP was significantly greater in patients without advanced fibrosis than with advanced fibrosis in our study. 

In the present study, we focused in the association between change in LSM and BW reduction due to *HSD17B13* genotype. After 1-year of diet therapy, the LSM significantly decreased from baseline and the change in LSM was positively associated with change in BW (data not shown). Change in LSM was not different between both *PNPLA3* and *HSD17B13* genotypes. Although the BW at baseline and change in BW were not different between *HSD17B13* AA and AG/GG, the change in LSM was significantly associated with change in BW only in patients with the AG/GG genotype not the AA genotype. This result means the reduction of BW is a good therapeutic target for improvement of liver fibrosis in patients with the AG/GG genotype, but not the AA genotype. Indeed, when the patients achieved BW reduction of more than 5% or 7%, recommended in guidelines, the reduction of LSM was significantly greater in patients with the AG/GG genotype than the AA genotype. We previously reported that carriage of the *HSD17B13* G allele attenuated the effect of the *PNPLA3* GG genotype in advanced liver fibrosis [16]. To our knowledge, no report has investigated the effect of the *HSD17B13* polymorphisms in response to BW reduction in Asian patients with NAFLD. The results of this study suggest that carriage of *HSD17B13* rs6834314 AA showed resistance to the therapeutic effects of BW reduction in the improvement of liver fibrosis. 

The *HSD17B13* was identified as a locus associated with elevated levels of ALT in a large-scale genome-wide association study and presumed to reflect hepatic fat accumulation [31]. *HSD17B13* encodes a hepatic lipid droplet protein; it was found upregulated in patients with NAFLD [32]. Ma et al. reported that *HSD17B13* rs6834314 was associated with the histological features of NAFLD, such as hepatic steatosis, inflammation, and ballooning [15]. They demonstrated that *HSD17B13* exhibits retinol dehydrogenase activity, and that the single amino acid mutation at rs6834314 is associated with loss of *HSD17B13* enzymatic activity despite normal protein expression and localization [15]. Retinoid metabolism may play a role in hepatic steatosis and liver injury in patients with NAFLD. Thus, the patients with *HSD17B13* G allele tend to accumulate IHTG, but have protective effects against liver injury. Peripheral lipolysis was identified as the main source of IHTG and hepatic steatosis [33,34]. The reduction in BW following lifestyle modification implies the reduction of lipolysis in peripheral tissues and of free fatty acid delivery to the liver. This beneficial effect might be shown more significantly in patients with the *HSD17B13* AG/GG genotype than the AA genotype.

This study has some limitations. It was a retrospective cohort and hospital-based study, and such studies are potentially subject to selection bias. Second, the lifestyle intervention in this study was not sufficient, in that the total amount of BW reduction was smaller than previous studies. However, only a few patients with NAFLD can achieve and keep BW reduction of more than 7%. This study reflects the reality of lifestyle intervention. Because of the small sample size, we could not evaluate the difference in ΔLSM and correlation with ΔBW in each fibrosis stages. The LSM is known to be affected by, e.g., serum level of AST, ALT, splenoportal dynamics, and obesity [20,35,36,37]. We could not exclude the possibility of these effects on LSM. The strengths of this study include the diagnostic method used and the relatively large number of subjects. Confirmation of the NAFLD diagnosis by liver biopsy reveals histological findings and avoids the uncertainty associated with ultrasonography for NAFLD diagnosis. 

## 5. Conclusions

In conclusion, the results of this cohort study suggest that the *HSD17B13* rs6834314 polymorphism is associated with change in LSM, which might affect liver fibrosis by lifestyle intervention. The incomplete response to diet therapy was thought to be due to inadequate BW reduction. We suggest that the target rate of BW reduction, in order to improve liver histology, needs to be set by considering genetic assessments. The approach, including genetic assessments, may contribute to the establishment of appropriate therapeutic strategies to treat NAFLD. 

## Figures and Tables

**Figure 1 genes-12-00628-f001:**
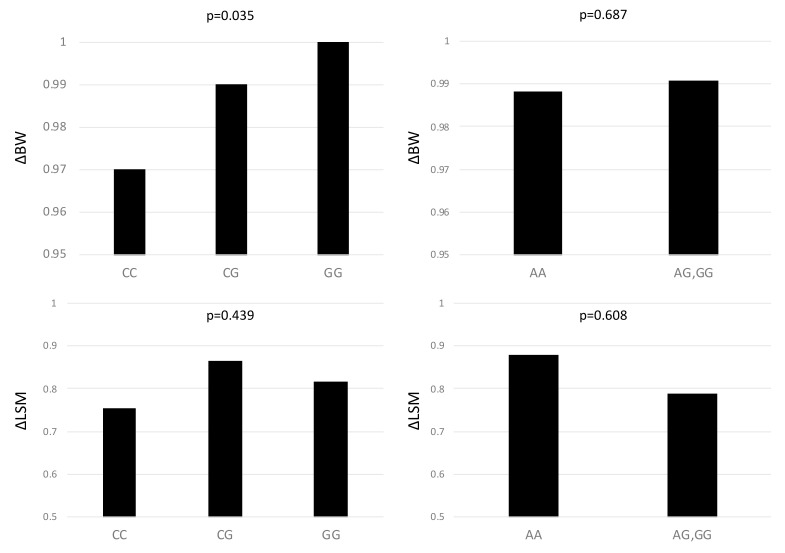
The changes in body weight and liver stiffness measurement according to *PNPLA3* rs738409 genotypes and *HSD17B13* rs6834314 in 140 patients with non-alcoholic fatty liver disease.

**Figure 2 genes-12-00628-f002:**
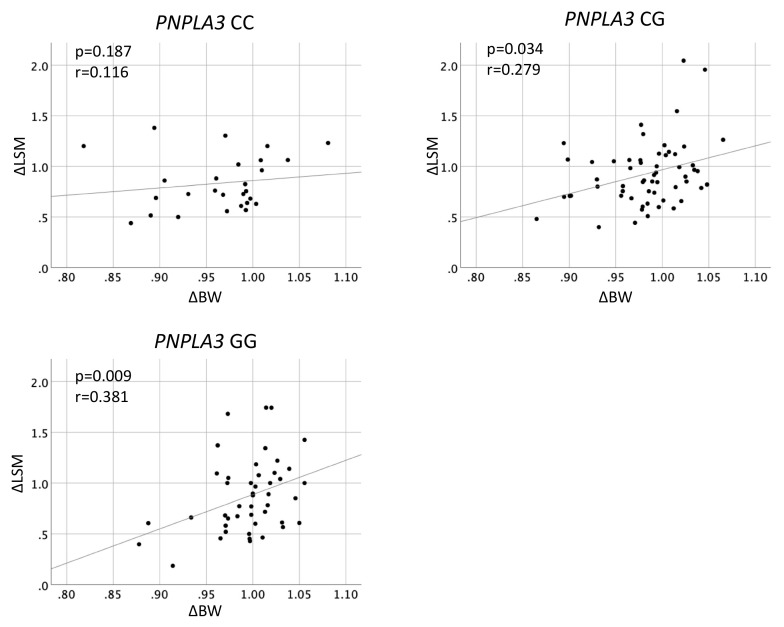
Correlation between changes in body weight and in liver stiffness measurement according to *PNPLA3* rs738409 genotypes in 140 patients with non-alcoholic fatty liver disease.

**Figure 3 genes-12-00628-f003:**
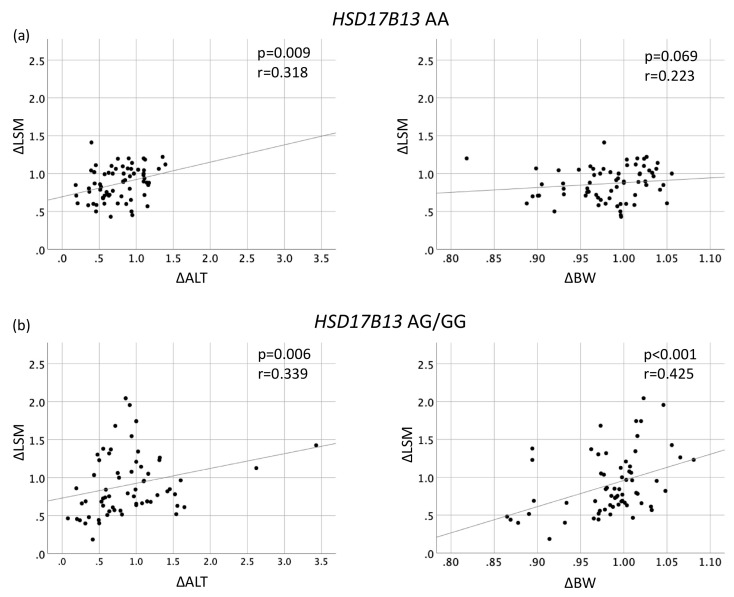
Correlation between changes in liver stiffness measurement and changes in serum ALT level, and in body weight in patients with (**a**) *HSD17B13* rs6834314 AA, (**b**) *HSD17B13* rs6834314 AG/GG.

**Figure 4 genes-12-00628-f004:**
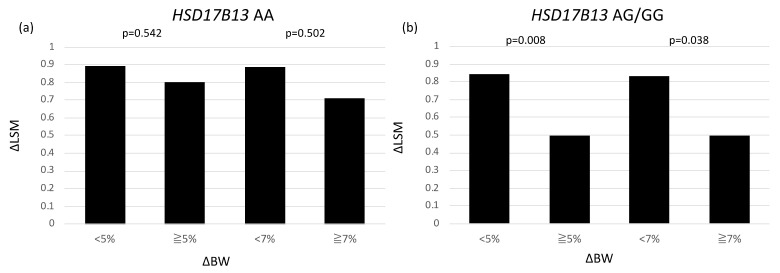
Change in live stiffness measurement, according to body weight reduction in patients with (**a**) HSD17B13 rs6834314 AA, (**b**) *HSD17B13* rs6834314 AG/GG.

**Table 1 genes-12-00628-t001:** Characteristics of the NAFLD patients according to the *PNPLA3* genotype.

Variable	*PNPLA3*			
	CC *n* = 30	CG *n* = 58	GG *n* = 52	*p*-Value
*HSD17B13*, AA/AG, GG	15/15	30/28	27/25	0.984
Sex, female	12 (40.0%)	31 (53.4%)	26 (50.0%)	0.485
Age, years	54 (23–81)	55 (19–75)	60 (17–76)	0.291
BMI, kg/m^2^	28.3 (23.3–45.2)	27.2 (20.8–44.0)	26.4 (17.6–41.0)	0.083
Hypertension	14 (46.7%)	25 (43.1%)	20 (38.5%)	0.429
Diabetes	15 (50.0%)	25 (43.1%)	23 (44.2%)	0.819
Hyperlipidemia	19 (63.3%)	36 (62.1%)	25 (48.1%)	0.248
Albumin, g/dL	4.5 (3.8–5.0)	4.5 (3.5–5.2)	4.4 (3.7–5.0)	0.825
AST, IU/L	32.5 (12–93)	42.5 (18–162)	47.5 (22–155)	0.186
ALT, IU/L	46 (18–146)	53.5 (19–233)	56 (14–166)	0.507
GGT, IU/L	71.5 (18–279)	59 (14–533)	59.5 (14–202)	0.833
Platelet count, ×10^3^/μL	239.5 (90–420)	226.5 (81–377)	213 (75–444)	0.210
Total cholesterol, mg/dL	195.5 (148–347)	205 (137–281)	201 (94–311)	0.548
TG, mg/dL	190 (55–721)	142.5 (49–739)	129 (53–644)	0.039
LDL–C, mg/dL	116 (33–252)	127 (59–219)	115 (36–177)	0.478
HDL–C, mg/dL	47 (21–67)	55 (30–93)	50 (29–96)	0.015
FPG, mg/dL	112 (77–232)	106.5 (76–172)	112 (71–236)	0.484
HbA1c, %	6.1 (5.2–11.2)	6.0 (5.0–8.2)	6.0 (5.2–8.7)	0.396
FIB-4 index	1.10 (0.28–7.83)	1.38 (0.33–6.43)	1.40 (0.39–7.73)	0.100
CAP, dB/m	323.5 (230–400)	308.5 (232–400)	299 (159–394)	0.052
LSM, kPa	8.9 (3.3–27.0)	8.6 (2.9–39.7)	9.6 (3.5–46.4)	0.778
Fibrosis stage (0/1/2/3/4)	15/9/4/2/0	19/23/6/5/5	18/13/7/10/4	0.276

Results are presented as *n* (%) for qualitative data or as medians for quantitative data. Abbreviations: *PNPLA3*, patatin-like phospholipase domain containing 3; BMI, body mass index; AST, aspartate aminotransferase; ALT, alanine aminotransferase; GGT, γ–glutamyl transferase; TG, triglyceride; LDL–C, low-density lipoprotein cholesterol; HDL–C, high-density lipoprotein cholesterol; FPG, fasting plasma glucose; FIB-4 index, Fibrosis-4 index.

**Table 2 genes-12-00628-t002:** Characteristics of the NAFLD patients according to the *HSD17B13* genotype.

Variable	*HSD17B13*		
	AA *n* = 72	AG/GG *n* = 68	*p*-Value
*PNPLA3*, CC/CG/GG	15/30/27	15/28/25	0.984
Sex, female	38 (52.8%)	31 (45.6%)	0.404
Age, years	55 (17–81)	54 (23–70)	0.925
BMI, kg/m^2^	27.0 (17.6–45.2)	28.2 (17.9–34.0)	0.809
Hypertension	31 (43.1%)	24 (35.3%)	0.865
Diabetes	33 (45.8%)	23 (33.8%)	0.866
Hyperlipidemia	45 (62.5%)	29 (42.6%)	0.232
Albumin, g/dL	4.4 (3.5–5.2)	4.5 (3.8–4.8)	0.306
AST, IU/L	50 (19–162)	38.5 (12–64)	0.002
ALT, IU/L	62 (21–233)	47.5 (14–165)	0.021
GGT, IU/L	66 (18–533)	56.5 (14–279)	0.316
Platelet count, ×10^3^/μL	213 (81–420)	237.5 (75–444)	0.060
Total cholesterol, mg/dL	195 (95–347)	204.5 (94–311)	0.108
TG, mg/dL	154 (49–721)	141 (53–739)	0.400
LDL–C, mg/dL	113.5 (36–252)	124 (33–219)	0.276
HDL–C, mg/dL	49 (21–82)	51 (30–96)	0.191
FPG, mg/dL	112 (71–222)	107 (77–236)	0.854
HbA1c, %	6.0 (5.2–8.5)	6.1 (5.0–11.2)	0.851
FIB-4 index	1.51 (0.33–7.83)	1.15 (0.28–7.73)	0.033
CAP, dB/m	313 (174–400)	304.5 (159–400)	0.370
LSM, kPa	9.4 (3.7–29.6)	8.1 (2.9–46.4)	0.158
Fibrosis stage (0/1/2/3/4)	20/26/11/10/5	32/19/6/7/4	0.210

Abbreviations are listed in the footnote to Table 1. *HSD17B13*, 17–β hydroxysteroid dehydrogenase 13.

**Table 3 genes-12-00628-t003:** The characteristics of 140 patients with nonalcoholic fatty liver disease at baseline and after 1 year diet therapy.

Variable	Baseline	After 1 Year	*p*-Value
*PNPLA3*, CC/CG/GG	30/58/52		
*HSD17B13*, AA/AG/GG	72/57/11		
BMI, kg/m^2^	27.6 (17.6–45.2)	26.7 (17.6–46.2)	<0.001
Albumin, g/dL	4.5 (3.5–5.2)	4.4 (3.6–5.3)	0.358
AST, IU/L	42 (12–162)	33 (14–121)	<0.001
ALT, IU/L	54 (14–233)	40 (8–216)	<0.001
GGT, IU/L	59.5 (14–533)	50.5 (9–456)	<0.001
Platelet count, ×10^3^/μL	222.5 (76–444)	227 (56–472)	0.124
Total cholesterol, mg/dL	201.5 (94–347)	193 (88–312)	0.493
TG, mg/dL	145 (49–739)	136 (49–714)	0.117
LDL–C, mg/dL	119 (33–252)	120 (33–231)	0.131
HDL–C, mg/dL	50 (21–96)	50.5 (22–116)	0.330
FPG, mg/dL	109 (71–236)	106.5 (61–387)	0.830
HbA1c, %	6.0 (5.0–11.2)	6.0 (4.9–11.8)	0.534
FIB-4 index	1.29 (0.28–7.83)	1.31 (0.27–11.58)	0.001
CAP, dB/m	309.5 (159–400)	298 (129–400)	0.020
LSM, kPa	8.8 (2.9–46.4)	6.5 (2.4–32.4)	<0.001

Abbreviations are listed in the footnotes of Table 1 and Table 2.

**Table 4 genes-12-00628-t004:** Factors associated with change in live stiffness measurement according to the *HSD17B13* genotype.

	Factors	β	95% CI	*p*-Value
*HSD17B13* AA	*PNPLA3* genotype	−0.058	−0.136 to −0.021	0.145
	Change in BW	0.454	−0.770 to −1.678	0.460
	Change in ALT	−0.111	−0.416 to −0.193	0.468
	Change in AST	0.194	−0.044 to −0.432	0.109
	Change in GGT	0.070	−0.012 to −0.152	0.094
	Change in TC	−0.087	−0.419 to −0.246	0.604
	Change in TG	0.083	−0.044 to −0.211	0.197
*HSD17B13* AG/GG	*PNPLA3* genotype	0.032	−0.107 to −0.172	0.645
	Change in BW	3.043	0.276 to −5.811	0.032
	Change in ALT	0.001	−0.509 to −0.510	0.998
	Change in AST	−0.050	−0.672 to −0.572	0.873
	Change in GGT	0.272	−0.181 to −0.725	0.233
	Change in TC	0.179	−0.434 to −0.793	0.560
	Change in TG	−0.121	−0.363 to −0.121	0.320

Abbreviations are listed in the footnote to Table 1 and Table 2.

## Data Availability

The data presented in this study are available upon request from the corresponding author. The data are not publicly available due to restriction by the institutional ethics committee.

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
