# Peer review of "The Effect of Genetic Polymorphism in Response to Body Weight Reduction in Japanese Patients with Nonalcoholic Fatty Liver Disease"

_genes, 2021, doi:10.3390/genes12050628_

Round 1

Reviewer 1 Report

The 140 cohort study found that Change in liver stiffness (LSM) was correlated to SNP of rs6834314 but not in PNPLA3. This is an interesting finding for NAFLD therapy.  The study is well design and well written.

Author Response

The 140 cohort study found that Change in liver stiffness (LSM) was correlated to SNP of rs6834314 but not in PNPLA3. This is an interesting finding for NAFLD therapy.  The study is well design and well written.

Response: We appreciate your comments. 

Reviewer 2 Report

The authors submitted a manuscript on the effect of genetic polymorphism on response to body weight and reduction in LSM. The manuscript is interesting, however there are some issues that need to be addressed before further processing.

  • Line 71-75: one of the aim is LSM changes, however the introduction gives little-to-none background on LSM in general and in NAFLD patients. This shoul be addressed. I suggest to cite and take inspiration from a recent editorial entitled "elastography: where are we now?" (DOI:  10.23736/S1121-421X.20.02773-7), also the authors should address the problem of LSM reliability in NAFLD (check and cite this recent paper on another MDPI journal: DOI:  10.3390/diagnostics10100795).
  • Line 94-95: from what reported in tables, the authors used the FIB-4 score, the authors should cite how this was calculated and its usefulness related to LSM (check and cite this recent article - DOI: https://doi.org/10.1016/j.aohep.2020.04.003)
  • Line 96-97: the authors give no information on how LSM was measured and reliability criteria, please add more information.
  • Line 123-130: were data tested for normality? Because the authors employ both parametric and none parametric tests. Also, for correlation tests the strength (weak, moderate, strong), etc. Also, it is not clear what the authors mean with LSM changes (reduction yes/no, reduction of a certain amount yes/no, quantitative reduction?)
  • Results: it is not clear why authors report differences in LSM between groups without considering fibrosis staging. It would be more accurate and logic to study difference in LSM F0vs F0, F1vsF1, etc. Also, according to the data, they appears to be scattered according to figure 2 and 3 - I suggest removing the images. 
  • Discussion: the authors should address reason behind changes in LSM in terms of transaminases (check and cite DOI: https://doi.org/10.3390/microorganisms8030348) and splenoportal dynamics (DOI1: 10.1007/s40477-020-00456-9  - DOI2: https://doi.org/10.1016/j.aohep.2020.07.004, and skin-to-liver distance (DOI:  10.3390/diagnostics10100795). 

Author Response

Response to Reviewer 2 Comments

The authors submitted a manuscript on the effect of genetic polymorphism on response to body weight and reduction in LSM. The manuscript is interesting, however there are some issues that need to be addressed before further processing.

Point 1: Line 71-75: one of the aim is LSM changes, however the introduction gives little-to-none background on LSM in general and in NAFLD patients. This shoul be addressed. I suggest to cite and take inspiration from a recent editorial entitled "elastography: where are we now?" (DOI:  10.23736/S1121-421X.20.02773-7), also the authors should address the problem of LSM reliability in NAFLD (check and cite this recent paper on another MDPI journal: DOI:  10.3390/diagnostics10100795).

Response 1: We appreciate your suggestion. We added the comment about LSM in “Introduction” section. (Line 71-77)

Point 2: Line 94-95: from what reported in tables, the authors used the FIB-4 score, the authors should cite how this was calculated and its usefulness related to LSM (check and cite this recent article - DOI: https://doi.org/10.1016/j.aohep.2020.04.003)

Line 96-97: the authors give no information on how LSM was measured and reliability criteria, please add more information.

Response 2: We appreciate your suggestion. We added the calculation of FIB-4 index and procedure of LSM in “material and method” section. (Line 103-111)

Point 3: Line 123-130: were data tested for normality? Because the authors employ both parametric and none parametric tests. Also, for correlation tests the strength (weak, moderate, strong), etc. Also, it is not clear what the authors mean with LSM changes (reduction yes/no, reduction of a certain amount yes/no, quantitative reduction?)

Response 3: We chose parametric or non-parametric analysis as appropriate. We added the correlation strength according to correlation coefficient. In this study, we analysed quantitative change in LSM. (Line 141-145)

Point 4: Results: it is not clear why authors report differences in LSM between groups without considering fibrosis staging. It would be more accurate and logic to study difference in LSM F0vs F0, F1vsF1, etc. Also, according to the data, they appears to be scattered according to figure 2 and 3 - I suggest removing the images.

Response 4: We focused on the difference in ΔLSM according to SNPs in Figure 1. As we showed in Tagle1 and Table 2, there was no significant difference in fibrosis stage between both SNPs. Furthermore, ΔLSM was not different in each fibrosis stage and SNPs. So, we did not mention the association between ΔLSM and fibrosis stage. We added the comment about this association in Line 187-188.

Because of small sample size of advanced fibrosis stage, we could not evaluate correlation between ΔLSM and ΔBW or ΔALT in each fibrosis stages (Figure 2, 3). We added these results in limitation. Line 314-316.

I might misunderstand your suggestion. Please tell us the detail of your comments. Which images should we remove?

Point 5: Discussion: the authors should address reason behind changes in LSM in terms of transaminases (check and cite DOI: https://doi.org/10.3390/microorganisms8030348) and splenoportal dynamics (DOI1: 10.1007/s40477-020-00456-9  - DOI2: https://doi.org/10.1016/j.aohep.2020.07.004, and skin-to-liver distance (DOI:  10.3390/diagnostics10100795).

Response 5: As reviewer pointed, ΔLSM was correlated with ΔALT in this study. Unfortunately, we could not evaluate the effects of Spleno-portal dynamics and obesity. We appreciate your suggestion. We added the possibility of these effects on LSM in limitation. (Line 316-318)

Round 2

Reviewer 2 Report

The authors have edited the manuscript according to reviewers' comments.

The manuscript, in its current form, it's acceptable for publication.